# ENHANCING MUTUAL INFORMATION ESTIMATION IN SELF-INTERPRETABLE GRAPH NEURAL NETWORKS

## ABSTRACT

Graph neural networks (GNNs) with self-interpretability are pivotal in various high-stakes and scientific domains. The information bottleneck (IB) principle holds promise to infuse GNNs with inherent interpretability. In particular, the graph information bottleneck (GIB) framework identifies key subgraphs from the input graph $G$ that have high mutual information (MI) with the predictions while maintaining minimum MI with $G$. The major challenge is dealing with irregular graph structures and gauging the conditional probabilities for evaluating MI between these subgraphs and $G$. Existing methods for estimating the MI between graphs often present distorted and loose estimations, thereby undermining model efficacy. In this work, we propose a novel framework GEMINI for training self-interpretable graph models, which tackles the key challenge of graph MI estimations. We construct a variational distribution over critical subgraphs, based on which an efficient MI upper bound estimator for graphs is built. Besides the proposed theoretical framework, we devise a practical instantiation of different modules in GEMINI. We compare GEMINI thoroughly with both self-interpretable GNNs and post-hoc explanation methods on eight datasets with both interpretation and prediction performance metrics. Results reveal that GEMINI outperforms state-of-the-art self-interpretable GNNs on interpretability and achieves comparable prediction performance compared with mainstream GNNs.

## 1 INTRODUCTION

Graph data permeates various domains, from social media platforms (Nettleton, 2013; Knoke & Yang, 2019; Xia et al., 2021) and transportation systems (Zhou et al., 2020; Wang et al., 2020) to chemical compounds (Gilmer et al., 2017; Wieder et al., 2020; Hao et al., 2020). In light of this, Graph neural networks (GNNs) have emerged as *de facto* models to encode graph-structured data due to their great power of synthesizing graph structure and features (Kipf & Welling, 2016; Hamilton et al., 2017; Veličković et al., 2017; Xu et al., 2018; Corso et al., 2020). Yet, beyond sheer predictive prowess, there is an increasing emphasis on model interpretability and transparency. For example, scientists identify important molecular functional groups or find some disorder-specific regions of interest in biomedical analysis (Wencel-Delord & Glorius, 2013; Li et al., 2021; Cui et al., 2022) with interpretable models. In the sphere of financial fraud detection, interpretable models are paramount; their clarity can prevent false alarms, thereby safeguarding both customer trust and corporate reputation (Rao et al., 2020; Yang et al., 2021; Wang et al., 2021).

GNNs achieve their expressiveness by the synthesis of graph structure and features. Yet, the synthesis complicates the endeavor to interpret and understand the underlying model behaviors. In response to such complexities, a considerable body of research has been devoted to developing post-hoc explainers for GNNs, i.e., try to find critical subgraphs that primarily dominate model predictions after training (Ying et al., 2019; Luo et al., 2020; Yuan et al., 2021; Shan et al., 2021; Xia et al., 2022). Nonetheless, these post-hoc methods tend to suffer from sensitivity and reliance problems (Adebayo et al., 2021). Given these challenges, there is a surging interest in architecting GNNs that possess self-interpretability. Such interpretability aligns more closely with a model's intrinsic reasoning processes, potentially bolstering its robustness as well. Central to these efforts is the incorporation of the Information Bottleneck (IB) theory (Tishby et al., 2000; Tishby & Zaslavsky, 2015; Alemi et al., 2016) into the foundational architecture and optimization objectives of

GNNs (Yu et al., 2021; 2020; Wu et al., 2020; Miao et al., 2022; Lee et al., 2023), i.e., the Graph Information Bottleneck (GIB) framework.

The IB theory relies on gauging the mutual information (MI) between observed data variables and models' learned intermediate variables. In the context of the GIB framework, the objective is to identify a critical subgraph $G_{\text{sub}}$ among the input graph $G$, such that $G_{\text{sub}}$ exhibits high MI with the prediction variable $Y$, while maintaining a low MI with $G$, i.e., a parameterized conditional distribution $p_\phi(G_{\text{sub}}|G)$ should be learned. Computing MI precisely for graph variables is challenging because of the difficulty in the calculation of $p_\phi(G_{\text{sub}}|G)$ for non-Euclidean graph data. Existing works largely overlooked the intrinsic subgraph matching problem when calculating $p_\phi(G_{\text{sub}}|G)$, which we show is critical for graph MI estimation. Some works establish an upper bound by introducing a prior marginal distribution, denoted as $r(G_{\text{sub}})$. This can be done either in the graph space (Wu et al., 2020; Miao et al., 2022) or in the graph embedding space (Sun et al., 2022). Then an upper bound of MI between $G_{\text{sub}}$ and $G$ is derived by computing the Kullback-Leibler (KL) divergence from the prior distribution $r(G_{\text{sub}})$ to the conditional distribution $p_\phi(G_{\text{sub}}|G)$. Optimally, $r(G_{\text{sub}})$ should align with $p(G_{\text{sub}})$, which is derived from $p(G)$ and $p_\phi(G_{\text{sub}}|G)$, such that the KL divergence equates the MI between $G_{\text{sub}}$ and $G$ (in expectation). However, the non-Euclidean nature of graphs and variability in $p_\phi(G_{\text{sub}}|G)$ make approximation challenging. Existing approaches implicitly ignore the subgraph matching problem when calculating $p_\phi(G_{\text{sub}}|G)$ and $r(G_{\text{sub}})$. This introduces a significant deviation from the ideal prior, yielding a poor estimation. Another way is to approximate the MI through the Donsker-Varadhan (DV) representation (Donsker & Varadhan, 1975) of the KL divergence. Nevertheless, these methods suffer from extremely high computational cost and numerical instability during the model optimization procedure (Yu et al., 2020; 2021).

As stated above, the key challenge in applying the IB theory to graph domains is the computation of the MI between $G_{\text{sub}}$ and $G$. To address the shortcomings of previous works, we introduce a novel approach called *GEMINI*, which achieves self-interpretable *G*NNs with an *E*nhanced *M*utual *IN*formation est*I*mator for both efficiency and effectiveness. This estimator is constructed through the following key steps. *Firstly*, we leverage a MI upper bound estimator relying solely on the conditional probability distribution $p_\phi(G_{\text{sub}}|G)$, which has been largely overlooked in existing GIB literature. In this way, the obstacle of designating a prior graph distribution $r(G_{\text{sub}})$ and calculating KL divergence between graph distributions could be circumvented. *Secondly*, computing the distribution $p_\phi(G_{\text{sub}}|G)$ itself is also nontrivial since it relates to the generation procedure of $G_{\text{sub}}$ and is hindered by the NP-hardness of the subgraph matching problem. Because when calculating the $p_\phi(G_{\text{sub}}|G)$ precisely based on learned structural probabilities, the position of $G_{\text{sub}}$ on $G$ should be determined. However, the canonical position of a specific $G_{\text{sub}}$ sample on $G$ is unknown without implicit and strict assumptions. Considering that, we develop a variational distribution $q_\theta(G_{\text{sub}}|G)$ as a substitute of $p_\phi(G_{\text{sub}}|G)$, as well as a specific training objective for $q_\theta$. *Thirdly*, the instantiation of variational distribution $q_\theta(G_{\text{sub}}|G)$ requires scrupulous attention to detail because the approximation quality of $q_\theta(G_{\text{sub}}|G)$ directly affects the quality of MI upper bound estimations. To build $q_\theta(G_{\text{sub}}|G)$, we propose to extract graph representations of $G_{\text{sub}}$ and $G$ and construct probability distributions in the representation space. During training, a generator determining $p_\phi(G_{\text{sub}}|G)$, $q_\theta(G_{\text{sub}}|G)$, and a predictor responsible for model's predictive capability will be optimized alternatively. Based on the new MI upper bound estimator, we efficiently and effectively incorporate the IB theory into the graph domain, minimizing MI estimation bias for better interpretation and prediction.

We extensively evaluate GEMINI on eight datasets for interpretation (Sec. 5.2) and prediction (Sec. 5.3 ) metrics and investigate the effect of coefficient and randomness in GEMINI (Sec. 5.4). Results reveal that GEMINI achieves a significant interpretation performance gain and remains a competitive prediction performance compared with state-of-the-art interpretable as well as mainstream GNN models. Interestingly, GEMINI could automatically learn to generate sparse critical subgraphs even without explicit sparsity constraints. Furthermore, the best coefficient of our proposed MI upper bound constraint varies across different datasets, which generally requires certain tuning.

## 2  NOTATIONS AND PRELIMINARIES

We first introduce key notations and concepts necessary. Let $G = (V, E)$ denote a graph with node set $V$ and edge set $E$. $G$ may contain node features $\{X_v \in \mathbb{R}^d | v \in V\}$ and edge features $\{X_e \in \mathbb{R}^d | v \in E\}$ with feature dimension $d$. We denote neighbors of node $i$ as $\mathcal{N}(i)$.

**The information bottleneck framework.** The IB principle is grounded in mutual information between random variables (Tishby et al., 2000). For random variables $x$ and $y$, the MI is defined as $I(x; y) = \mathbb{E}_{p(x,y)} \left[ \log \frac{p(y|x)}{p(y)} \right]$. Generally, $x$ represents the input feature random variable and $y$ is the label variable related to $x$. Given the data distribution $p(x, y)$, the IB framework aims to learn a **conditional distribution** $p_\phi(z|x)$ of an intermediate variable $z$, which has the minimal sufficient information in $x$ when inferring $y$. $p_\phi(z|x)$ is generally determined by a **parameterized network** $g_\phi$ accompanied by a specific **generation (or sampling) procedure** [1]. The optimization objective of IB could be formulated in Eq. 1.

$$\underset{p_\phi(z|x)}{\arg \min} -I(z; y) + \beta I(x; z) \tag{1}$$

where $\beta$ is a hyper-parameter to trade off predictive performance against robustness and interpretability. However, the exact calculation of MI is often prohibitive for IB in Eq. 1. Constructing upper or lower bounds of these terms becomes a practical solution. For example, one can obtain a widely adopted variational upper bound of $I(x; z)$ by introducing a prior distribution $r(z)$ (Alemi et al., 2016): $I(x; z) \leq \mathbb{E}_{p(x)} \left[ KL(p_\phi(z|x) \| r(z)) \right]$.

**Definitions of distributions.** We adopt $p$ to represent *groundtruth* distributions [2] and $q$ to denote variational distributions. Different subscripts of $p$ and $q$ indicate different networks. As shown in Eq. 1 and Eq. 2, we denote $p_\phi(z|x)$ (or $p_\phi(G_{\text{sub}}|G)$ ) as the conditional probability distribution of $z$ (or $G_{\text{sub}}$) constructed by $g_\phi$ and the generation procedure. Once $p_\phi(z|x)$ is clarified, the joint distribution $p(x, z)$ and the marginal distribution $p(z)$ are **determined**, i.e., $p(x, z) \triangleq p(x)p_\phi(z|x)$, $p(z) \triangleq \int_x p_\phi(z|x)dx$. Further, the conditional probability $p(y|z)$ is also **settled** correspondingly according to the Bayes' theorem and the data distribution $p(x, y)$, i.e., $p(y|z) \triangleq \int_x p(x|z)p(y|x) \triangleq \int_x \frac{p(x)p_\phi(z|x)}{p(z)}p(y|x)$. *Note that $x$ and $z$ in the above distributions could be substituted with $G$ and $G_{sub}$ respectively when considering the GIB framework.*

**Graph information bottleneck.** To improve GNNs' interpretability and robustness, recent works (Wu et al., 2020; Yu et al., 2020) apply the IB principle in graph learning by instantiating $z$ in IB as a critical subgraph $G_{\text{sub}}$ and $x$ as the input graph $G$ from which $G_{\text{sub}}$ is generated. Hence, the optimization objective of the GIB framework is

$$\underset{p_\phi(G_{\text{sub}}|G)}{\arg \min} -I(G_{\text{sub}}; Y) + \beta I(G; G_{\text{sub}}) \tag{2}$$

where $G$ and $Y$ are observed graph data and the corresponding graph label signal, respectively. $G_{\text{sub}}$ is regarded as self-interpretation of model's prediction. We adopt $G_{\text{sub}}, G$ to represent graph random variables and $g_{\text{sub}}, g$ to indicate concrete instances. For rigorous definition of the graph random variables and the probability space, please refer to Appendix A.1.

# 3 LIMITATION ANALYSIS OF EXISTING GRAPH MI ESTIMATIONS

In this section, we elaborate on the limitations of existing graph MI estimation methods. The most widely adopted assumption is that the existence of edges is independent in the graph. Thus, previous works first learn edge probabilities and then sample each edge independently to generate a subgraph (Miao et al., 2022). $p_\phi(g_{\text{sub}}|g)$ is the probability that $g_{\text{sub}}$ is a generated subgraph by $g$ based on an underlying subgraph generation process parameterized by $\phi$. Existing implementations regard $g_{\text{sub}}$ as a 0-1 edge-weighted variant of $g$, where $g_{\text{sub}}$ and $g$ possess the same node set, as depicted in the

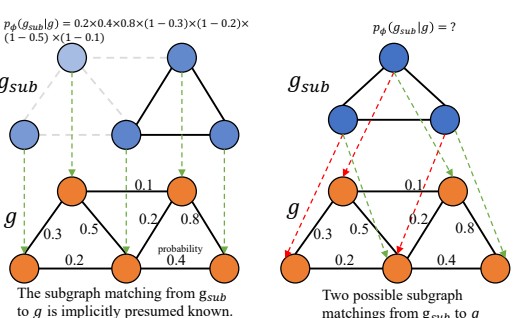

Figure 1: Subgraph matching in $p_\phi(g_{\text{sub}}|g)$.

---

[1]Typically, $g_\phi$ is responsible for distribution parameters of $p_\phi(z|x)$, and the sampling process contains no learnable parameters.

[2]The groundtruth means distributions determined by the generator $p_\phi(z)$ and $p(x, y)$ in the IB framework.

left part of Figure 1. Thus, the value of $p_\phi(g_{sub}|g)$ is the product of edges probabilities in $g_{sub}$. This formulation implicitly assumes that the subgraph matching (a mapping between nodes that preserves edge relations) is **unique and known**, as shown by the green arrows in the left part of Figure 1.

However, we argue that the subgraph matching between $g_{sub}$ and $g$ can be **non-unique** when $g_{sub}$ has fewer nodes than $g$ (since nodes can also be sampled), as indicated in the right part of Figure 1. Even if $g_{sub}$ and $g$ has identical node set, the subgraph matching can be non-unique as well. Calculating the exact value of $p_\phi(g_{sub}|g)$ needs to find all possible subgraph matchings between $g_{sub}$ and $g$, which is impractical since subgraph matching for general graphs is NP-hard (Lewis, 1983). Moreover, established MI upper bounds generally rely on a prior $r(G_{sub})$. Computing the prior distribution $r(G_{sub})$ over graphs is equally challenging as $p_\phi(G_{sub}|G)$. To conclude, existing methods largely overlooked the intrinsic subgraph matching difficulty when calculating $p_\phi(G_{sub}|g)$ for graph MI estimation, hence resulting in distorted estimations. Please refer to Appendix A.1 for a rigorous formulation of $p_\phi(G_{sub}|G)$.

## 4 THE GEMINI FRAMEWORK

### 4.1 THEORETICAL MOTIVATION AND FORMULATION OF GEMINI

In the following, we delve into our treatments for $I(G_{sub}; Y)$ and $I(G; G_{sub})$ in GEMINI.

**The predictive capability term $I(G_{\mathbf{sub}}; Y)$.** The first term $I(G_{sub}; Y)$ of GIB in Eq. 2 measures the correlation between learned critical subgraph $G_{sub}$ and prediction signal $Y$, i.e., model's predictive capability with $G_{sub}$. As stated in Sec. 2, once $p_\phi(G_{sub}|G)$ is settled, $p(G_{sub})$ and $p(Y|G_{sub})$ are determined, hence the MI term $I(G_{sub}; Y)$ could be calculated deterministically. However, because of the integral structure in $p(G_{sub})$ and $p(Y|G_{sub})$, directly calculating $I(G_{sub}; Y)$ is nontrivial. Hence, we adopt a variational distribution $q_\omega(Y|G_{sub})$ as a proxy for $p(Y|G_{sub})$ and induce an lower bound of $I(G_{sub}; Y)$ correspondingly.

$$
\begin{aligned}
I(G_{sub}; Y) &= \mathbb{E}_{p(G_{sub},Y)}\left[\log \frac{q_\omega(Y|G_{sub})}{p(Y)}\right] + \mathbb{E}_{p(G_{sub})}\left[KL(p(Y|G_{sub})\|q_\omega(Y|G_{sub}))\right] \\
&\geq \mathbb{E}_{p(G_{sub},Y)}\left[\log \frac{q_\omega(Y|G_{sub})}{p(Y)}\right] = \mathbb{E}_{p(G_{sub},Y)}\left[\log q_\omega(Y|G_{sub})\right] + H(Y)
\end{aligned}
\tag{3}
$$

where $H(Y)$ indicates the entropy of label $Y$. Since $H(Y)$ is determined by the data distribution and does not depend on $q_\omega$, we thus focus on $\mathbb{E}_{p(G_{sub},Y)}\left[\log q_\omega(Y|G_{sub})\right]$ when optimizing $q_\omega$. When adopting mini-batch samples, we define the loss $L_Y$ as in Eq. 4.

$$
L_Y = \frac{1}{K}\sum_{i=1}^{K}\left[\log q_\omega(Y^{(i)}|G_{sub}^{(i)})\right]
\tag{4}
$$

which is responsible for the model's predictive performance.

**The MI constraint term $I(G; G_{\mathbf{sub}})$.** In light of the analysis of Sec. 3, we introduce a **variational distribution** $q_\theta(G_{\mathbf{sub}}|G)$, which operates as a proxy of $p_\phi(G_{sub}|G)$ to address the challenges as described in Sec. 3. Specifically, we construct $q_\theta(G_{sub}|G)$ such that it could be computed efficiently for arbitrary $G_{sub}$ and $G$ samples. Moreover, we introduce variational upper bounds of MI $I(G; G_{sub})$ based solely on the constructed variational $q_\theta(G_{sub}|G)$, instead of relying on a prior $r(G_{sub})$, which can also be tricky and difficult to implement. We defer the detailed instantiation of $q_\theta(G_{sub}|G)$ to Sec. 4.2.

Once a practical $q_\theta$ is constructed, we could adopt sample-based MI variational upper bounds based merely on $q_\theta$, avoiding directly tackling with a specific prior $r(G_{sub})$. For example, we could upper-bound $I(G; G_{sub})$ using the MI upper bound CLUB (Cheng et al., 2020) and adapt it for graphs:

$$
I(G; G_{sub}) \leq \mathbb{E}_{p(G;G_{sub})}\left[\log p_\phi(G_{sub}|G)\right] - \mathbb{E}_{p(G)}\mathbb{E}_{p(G_{sub})}\left[\log p_\phi(G_{sub}|G)\right]
\tag{5}
$$

Note that the ground truth distribution $p_\phi(G_{sub}|G)$ is required in Eq. 5. By substituting $p_\phi(G_{sub}|G)$ with $q_\theta(G_{sub}|G)$ and adopting mini-batch samples, we could construct the graph-based CLUB (GCLUB) MI upper bound estimator:

$$
\tilde{I}_{\text{GCLUB}}(G; G_{sub}) \triangleq \frac{1}{K}\sum_{i=1}^{K}\left[\log q_\theta(G_{sub}^i|G^i) - \frac{1}{K}\sum_{j}^{K}\log q_\theta(G_{sub}^j|G^i)\right] \triangleq L_{\text{GCLUB}}
\tag{6}
$$

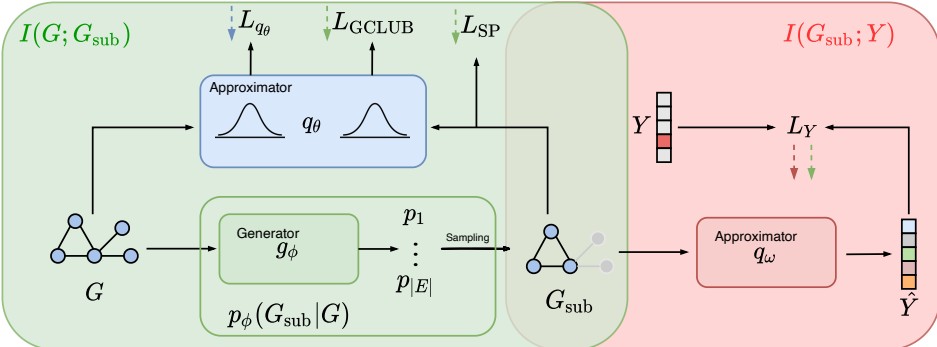

Figure 2: The framework of GEMINI which mainly consists of three modules $g_\phi$, $q_\theta$, and $q_\omega$. Solid arrows are computation flows, while colored dotted arrows next to losses represent gradient directions, indicating that modules with the same bounding color will be affected by specific losses.

The bound in Eq. 6 only relies on sampled $(G, G_{\text{sub}})$ pairs and the variational distribution $q_\theta$, leading to convenient and efficient calculations in practical implementation. Note that we also define the loss $L_{\text{GCLUB}}$ as $\tilde{I}_{\text{GCLUB}}(G; G_{\text{sub}})$ for mini-batch samples. It is worth clarifying that the form of upper bound estimator remains flexible in our GEMINI framework. One can directly plug in another estimator which solely relies on $p_\phi(G_{\text{sub}}|G)$ and substitute $p_\phi$ with our $q_\theta$ in GEMINI, for example, the leave-one-out (LOO) MI upper bound estimator (Poole et al., 2019).

Since the bound in Eq. 6 is constructed based on the variational $q_\theta(G_{\text{sub}}|G)$ which is different from the ground truth $p_\phi(G_{\text{sub}}|G)$, $q_\theta$ requires specific optimization to be a reasonable substitute of $p_\phi(G_{\text{sub}}|G)$ and ensure the effectiveness of $\tilde{I}_{\text{GCLUB}}(G; G_{\text{sub}})$. In the following, we will go into detail about the optimization objective and training scheme of $q_\theta$.

**The optimization of variational distribution** $q_\theta(G_{\textbf{sub}}|G)$**.** The goal of training $q_\theta$ is to make reasonable variational bounds when substituting $p_\phi$ by $q_\theta$ in MI upper bound estimators in GEMINI. Hence, we hope that the MI $I(G; G_{\text{sub}})$ determined by $p_\phi$ could be properly approximated by $q_\theta$, i.e., $I(G; G_{\text{sub}}) \approx \mathbb{E}_{p(G, G_{\text{sub}})}\left[\frac{q_\theta(G_{\text{sub}}|G)}{p(G_{\text{sub}})}\right]$, meaning $q_\theta$ is a *good* substitute for $p_\phi$. We could derive an optimization objective for $q_\theta$ based on similar techniques as in Eq. 3

$$
\begin{aligned}
I(G; G_{\text{sub}}) &= \mathbb{E}_{p(G, G_{\text{sub}})}\left[\log \frac{q_\theta(G_{\text{sub}}|G)}{p(G_{\text{sub}})}\right] + \mathbb{E}_{p(G)}\left[KL(p_\phi(G_{\text{sub}}|G)\|q_\theta(G_{\text{sub}}|G))\right] \\
&= \mathbb{E}_{p(G, G_{\text{sub}})}\left[\log q_\theta(G_{\text{sub}}|G)\right] + H(G_{\text{sub}}) + \mathbb{E}_{p(G)}\left[KL(p_\phi(G_{\text{sub}}|G)\|q_\theta(G_{\text{sub}}|G))\right]
\end{aligned}
\tag{7}
$$

Eq. 7 indicates that maximizing $\mathbb{E}_{p(G, G_{\text{sub}})}\left[\log q_\theta(G_{\text{sub}}|G)\right]$ w.r.t $\theta$ equals to minimizing the KL divergence between $p_\phi(G_{\text{sub}}|G)$ and $q_\theta(G_{\text{sub}}|G)$. However, in practical implementation, $q_\theta$ is trained using mini-batch samples, i.e., $\frac{1}{K}\sum_{i=1}^{K}\log q_\theta(G_{\text{sub}}^{(i)}|G^{(i)})$. Because of the enormous and discrete characteristic of graph space, such an objective will result in large variances during the training of $q_\theta$. Inspired by MI lower bound estimation techniques (Poole et al., 2019), we adopt another scheme to optimize $q_\theta$ using the following objective:

$$
L_{q_\theta} \triangleq -\frac{1}{K}\sum_{i=1}^{K}\log \frac{q_\theta(G_{\text{sub}}^i|G^i)}{\frac{1}{K}\sum_{i=1}^{K} q_\theta(G_{\text{sub}}^i|G^j)}
\tag{8}
$$

$$
\arg\min_\theta \quad \mathbb{E}_{p(G^1, G_{\text{sub}}^1) \times \cdots \times p(G^K, G_{\text{sub}}^K)}\left[L_{q_\theta}\right]
\tag{9}
$$

where $G^2, \cdots, G^K$ are $K-1$ auxiliary graph random variables. The objective in Eq. 9 reduces the variance and has the optimal point at $q_\theta(G_{\text{sub}}|G) = p_\phi(G_{\text{sub}}|G)$ (Poole et al., 2019; Ma & Collins, 2018). Hence, optimizing $L_{q_\theta}$ w.r.t $\theta$ ensures the optimality of $\theta$ and improves the training stability, which is adopted in our practical implementation of GEMINI. We defer the overall training scheme of GEMINI in Algorithm 1 in Appendix A.4.

### 4.2 INSTANTIATION OF GEMINI

In this subsection, we elaborate on detailed instantiations of the GEMINI framework. $g_\phi$, $q_\theta$, and $q_\omega$ are instantiated as independent networks containing GNN encoders and extra modules. We depict the framework of GEMINI in Figure 2.

**Instantiation of $g_\phi$.** Since $G_{\text{sub}}$ should be a critical interpretable subgraph contained in the input graph, we formulate $G_{\text{sub}}$ as one generated by sampling nodes and edges on the input graph $G$. The sampling probability is learned by the generator $g_\phi$. After all probabilities are obtained, we sample each edge and node independently to formulate the critical subgraph $G_{\text{sub}}$. Specifically, we calculate edge probabilities using both nodes' representations, and induce node probabilities based on edge probabilities, as in Eq. 10. The node representation $z_i$ is extracted from a specific GNN encoder, e.g., GCN (Kipf & Welling, 2016) or GIN (Xu et al., 2018).

$$
\begin{aligned}
z_i, z_j &= \text{GNN}(X, E) \\
\hat{e}_{ij} &= \text{Sigmoid}(\text{MLP}([\text{mean}(z_i, z_j) \| \max(z_i, z_j) \| \min(z_i, z_j)])) \\
\hat{n}_i &= \max\{\hat{e}_{ij} | j \in \mathcal{N}(i)\}
\end{aligned}
\tag{10}
$$

In order to backpropagate gradients during the sampling, we adopt the Gumbel-Softmax trick (Jang et al., 2016) in the forward computation. The combination of sampled values and the original graph is regarded as an instance of the critical subgraph $G_{\text{sub}}$. Moreover, in order to support the instantiation of subsequent modules $q_\theta$ and $q_\omega$, we alter the message-passing procedure correspondingly which calculating representation of $G_{\text{sub}}$, as indicated in Eq. 11.

$$
\begin{aligned}
m_{ij}^k &= \text{Msg}(\{\hat{n}_i \odot X_i^k, \hat{n}_j \odot X_j^k\} \cup \{\hat{e}_{ij} \odot X_{ij}^k\}) \\
X_i^{k+1} &= \text{Agg}(\{\hat{n}_i \odot X_i^k\} \cup \{\hat{e}_{ij} \odot m_{ij}^k | j \in \mathcal{N}_i\})
\end{aligned}
\tag{11}
$$

where $\odot$ is element-wise multiplication, which applies sampled weights to the message generation and aggregation of a GNN encoder.

**Instantiation of $q_\theta$.** Since $q_\theta(G_{\text{sub}}|G)$ takes two graphs as input and should approximate a probability, we propose to model each graph as a Gaussian distribution in the embedding space.

$$
\begin{aligned}
z &= \text{GNN}_1(G), \quad \mu, \sigma = \text{Mean}_1(z), \text{Var}_1(z) \\
z_{\text{sub}} &= \text{GNN}_2(G_{\text{sub}}), \quad \mu_{\text{sub}} = \text{Mean}_2(z_{\text{sub}})
\end{aligned}
\tag{12}
$$

Once we obtain $\mu$, $\sigma$ and $\mu_{\text{sub}}$, and $q_\theta(G_{\text{sub}}|G)$ is formulated as $q_\theta(G_{\text{sub}}|G) \triangleq \mathcal{N}(\mu_{\text{sub}}; \mu, \sigma)$. Instead of predicting an independent value for each $(G, G_{\text{sub}})$ pair, the Gaussian formulation facilitates probability computation for batched graph samples, as we only need to predict each graph's mean and variance and calculate the cross-pairs' conditional probabilities in a closed-form manner.

**Instantiation of $q_\omega$.** The instantiation of $q_\omega(Y|G_{\text{sub}})$ is rather straightforward. We adopt a GNN encoder combined with an MLP module to output predictions, i.e., $\hat{Y} = \text{MLP}(\text{GNN}(G_{\text{sub}}))$.

Moreover, we add another information constraint $L_{sp}$ accompanied by the objective in Eq. 6 during training $g_\theta$. The construction of $L_{sp}$ is $L_{sp} = \sum_{e \in E} KL(Bern(\hat{e}) \| Bern(\tau))$, where $Bern(\tau)$ indicates the Bernoulli distribution with parameter $\tau$. $L_{sp}$ is essentially a sparsity constraint, as in most scenarios a sparse subgraph is expected for interpretation. The sparsity objective can also be approximately regarded as an MI constraint under some specific assumptions and simplifications (Miao et al., 2022). Therefore, the overall optimization objective for $g_\phi$ is:

$$
L_{g_\phi} = -L_Y + \lambda L_{GCLUB} + \gamma L_{sp}
\tag{13}
$$

where $\lambda$ and $\gamma$ are hyper-parameters. Based on the derived training objectives and instantiations above, we summarize the training procedure of GEMINI in Algorithm 1.

## 5 EXPERIMENTS

In this section, we evaluate the proposed instantiation of GEMINI. We mainly investigate prediction and interpretation performance, which is followed by analysis of different MI regularizations.

## 5.1 Experimental settings

**Datasets.** We adopt eight datasets for the evaluation: (1) MUTAG (Debnath et al., 1991), (2) MNIST-75sp (Knyazev et al., 2019), (3) MOLHIV (Wu et al., 2018; Hu et al., 2020), (4) Graph-SST2 (Yuan et al., 2020), (5-8) Spurious-Motif (Ying et al., 2019; Wu et al., 2022). Note that in the Spurious-Motif dataset, there is a parameter $b$ controlling the strength of spurious correlation within the dataset. We denote this dataset with parameter $b$ as SPMotif ($b$) and set $b$ to 0.3, 0.5, 0.7, 0.9 to form four datasets. We defer the details of these datasets in Appendix A.5.

**Baselines.** Since our framework concerns both interpretation and prediction performance, we compare it with graph interpretation and prediction models. *For interpretation evaluation*, We take both post-hoc GNN interpretation methods and self-interpretable methods as baselines. For post-hoc methods, baselines include GNNExplainer (Ying et al., 2019) and PGExplainer (Luo et al., 2020). For self-interpretable methods, baselines include GSAT (Miao et al., 2022) and DIR (Wu et al., 2022). *For prediction evaluation*, we take both backbone models and self-interpretable models as prediction performance baselines. Backbone models include GIN (Xu et al., 2018) and PNA (Corso et al., 2020). Self-interpretable models are GSAT (Miao et al., 2022) and DIR (Wu et al., 2022).

**Setup.** We compare GEMINI with baselines with both interpretation and prediction metrics. For GEMINI, we adopt GIN (Xu et al., 2018) and PNA (Corso et al., 2020) as backbones, following the setup in Miao et al. (2022). For the GSAT baseline, we adopt the same backbones. $\lambda$ is set to 0.05 and $\gamma$ is set to 1.0 by default. We use the original hyper-parameters for other baselines. We adopt the ACC for prediction and AUC for interpretation metrics respectively. Note that the interpretation performance is calculated based on learned edge/node probabilities and groundtruth edge/node interpretation labels (if exist). We repeat all experiments three times to report mean values and standard deviations. The best results are in bold and the second best are underlined.

Table 1: Interpretation performance comparison.

|  | MUTAG | MNIST-75sp | SPMotif (0.3) | SPMotif (0.5) | SPMotif (0.7) | SPMotif (0.9) |
|---|---|---|---|---|---|---|
| GNNExplainer | 61.77±0.95 | 52.88±0.11 | 50.55±0.10 | 44.25±0.53 | 47.72±0.34 | 46.36±0.23 |
| PGExplainer | 60.34±0.12 | 59.40±0.17 | 60.79±0.42 | 34.69±0.59 | 32.30±0.34 | 31.40±0.53 |
| DIR | 53.04±0.17 | 60.61±0.13 | 74.75±0.11 | 74.91±0.12 | 73.70 ±0.18 | 76.56±0.12 |
| GSAT-PNA | 99.49±0.12 | **63.40**±7.14 | 67.03±14.27 | 62.68±9.24 | 48.52±6.09 | 68.18±12.15 |
| GSAT-GIN | 98.37±0.47 | 60.14±1.14 | 76.44±11.7 | 81.16±2.30 | 83.01±0.55 | 76.20±8.39 |
| OURS-PNA | **99.54**±0.24 | 59.42±8.39 | 56.17±4.10 | 64.28±13.90 | 65.68±11.19 | 65.05±12.47 |
| OURS-GIN | 99.10±0.33 | 61.45±4.00 | **81.09**±3.66 | **82.40**±0.16 | **88.58**±8.92 | **79.65**±3.68 |

Table 2: Prediction performance comparison. AUC is adopted for MOLHIV for class imbalance.

|  | MOLHIV (AUC) | Graph-SST2 | MINIST-75sp | SPMotif (0.3) | SPMotif (0.5) | SPMotif (0.7) |
|---|---|---|---|---|---|---|
| GIN | **76.82**±0.38 | 81.91±0.54 | 94.36±0.31 | **82.74**±2.07 | 84.56±0.84 | 81.86±0.76 |
| PNA | 75.01±0.99 | 80.85±0.36 | 88.02±1.44 | 66.37±2.46 | 69.41±2.61 | 65.79±2.47 |
| DIR | 76.34±1.01 | 82.32±0.85 | 88.51±2.57 | 34.70±0.16 | 33.53±0.32 | 32.90±0.81 |
| GSAT-PNA | 70.88±0.53 | **83.83**±0.25 | 90.38±2.66 | 74.59±14.69 | 70.51±11.75 | 57.74±4.64 |
| GSAT-GIN | 74.23±1.44 | 83.70±0.17 | **96.68**±0.17 | 68.56±11.1 | 81.21±5.50 | 80.57±9.10 |
| OURS-PNA | 70.01±3.34 | 81.74±1.56 | 89.65±3.68 | 69.61±11.47 | 72.71±19.42 | 80.06±12.62 |
| OURS-GIN | 73.88±2.96 | 83.37±1.20 | 95.57±0.53 | 76.21±14.1 | **92.01**±2.24 | **89.19**±4.24 |

## 5.2 Interpretation performance

We list interpretation performance comparison in Table 1. The results reveal that GEMINI could achieve the best interpretation performance in most datasets, except the MNIST-75sp dataset. The largest improvement over the self-interpretable GSAT is about 11% on the SPMotif(0.9) dataset. In most datasets, GSAT achieves the second-best interpretation performance. If we consider only post-hoc methods, the improvement is even more significant, e.g., up to $40 \sim 50\%$ on spurious motif datasets. The reason is that the explanation quality of post-hoc explainers is susceptible to not only graph instances but also trained models. DIR is worse than our method and the GSAT baseline (with the GIN backbone). The results indicate that learning causal components and the environmental intervention simultaneously in the DIR framework may be challenging and volatile. Moreover, we notice the PNA backbone is worse than the GIN backbone generally. We consider that the reason comes from the removal of scalars used in PNA in our adaptation, which is necessary in our framework but may degrade the performance of PNA architecture.

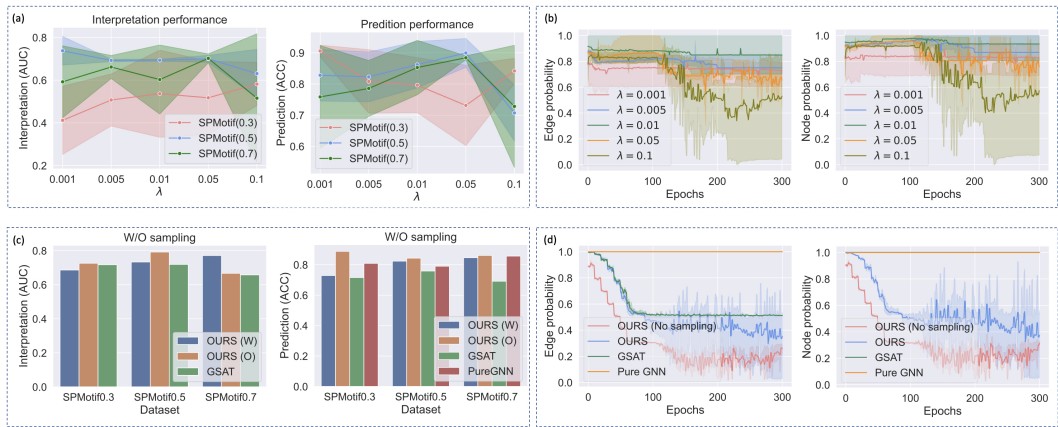

Figure 3: (a) The comparison of interpretation and prediction performance of GEMINI under different coefficients $\lambda$. (b) The comparison of edge and node probabilities on the validation set during model training (SPMotif0.7) for varying $\lambda$. (c) The comparison of interpretation and prediction performance of GEMINI under W/O sampling settings. (d) The comparison of edge and node probabilities during model training on the validation set (SPMotif0.7) under W/O sampling settings. Note that GSAT solely learns edge probabilities and node probabilities remain 1. Hence the curve of GIN overlaps over GSAT.

## 5.3 Prediction performance

As shown in Table 2, the interpretability could also improve the prediction performance in most cases. The proposed method achieves the best result or the second-best performance in four datasets and comparable results in other datasets. Compared with the self-interpretable GSAT, the improvement of GEMINI is more significant on spurious correlation datasets e.g., $\sim 9\%$ for the GIN backbone. Compared with pure GNN backbones, GEMINI could also enhance the prediction performance in most datasets, e.g., on SPMotif(0.5), the improvement is $\sim 8\%$ for GIN and $\sim 3\%$ for PNA. Note that the GIN backbone could also achieve competitive performance in several datasets, especially the MOLHIV, for which the class imbalance is severe, and subtle perturbation of the graph could degrade the model's prediction capability. For the invariant learning baseline, DIR performs worse even than GIN on the spurious correlation datasets, indicating that it may not separate causal and spurious parts well on these graphs. Moreover, both GEMINI and GSAT have better performance when adopting the GIN as backbone compared with the PNA backbone. These differences are generally consistent with results in Table 1. Hence, a proper GNN backbone for calculating graph representations is also significant for interpretation and prediction performance.

## 5.4 Analysis

**Effect of the $L_{GCLUB}$ term.** To investigate the effect of the IB upper bound loss $L_{GCLUB}$ more thoroughly, we remove the $L_{sp}$ term and report model performance under different coefficients of $L_{GCLUB}$ on the Spurious-Motif datasets. The interpretation and prediction performance is reported in Fig. 3(a). Further, we record the learned average edge and node probabilities on the validation set of SPMotif (0.7), which is shown in Fig. 3(b) (the curves are similar on other SPMotif datasets). We can conclude that a proper setting of $\lambda$ could improve both model's prediction and interpretation performance, e.g., 0.05 for SPMotif (0.5) and SPMotif (0.7). With the increase of $\lambda$ in the range [0.001, 0.05], the model's interpretation and prediction performance improves on SPMotif (0.5). For SPMotif (0.7), the interpretation performance remains generally unchanged while prediction performance improves. The prediction performance degrades for $\lambda$ in the range [0.001, 0.05], which may be caused by the relatively large information constraint since the spurious correlation is weaker than the other two datasets. Moreover, a too-large coefficient of $L_{GCLUB}$ may degrade the model's performance, for example, with the $\lambda$ set to 0.1, the model's interpretation and prediction decreases on both SPMotif (0.5) and SPMotif (0.7) dataset. We argue that as a mutual information regularizer, a large $L_{GCLUB}$ will impose strong constraints on the representations model learned, i.e., losing more

information about the original data and label signal, which may be harmful to both interpretation and prediction. Hence, a proper parameter searching for $\lambda$ is necessary for different datasets.

**Effect of the sampling procedure.** In this part, we aim to investigate the effect of Gumbel-Softmax sampling for $G_{\text{sub}}$ as well as learned subgraph probability distributions. We compare GEMINI w/o the sampling procedure with GSAT and a pure GNN (GIN) on the Spurious-Motif datasets. For the no-sampling version of GEMINI, we regard the learned edge/node probabilities as weights in message passing procedures. The performance comparison is illustrated in Fig. 3(c), and edge and node probabilities during training are demonstrated in Fig. 3(d) (on the validation set of SPMotif (0.7)). From Fig. 3(c), we can find that the no-sampling version of GEMINI achievers better prediction performance in all cases, and better interpretation performance in SPMotif (0.3) and SPMotif (0.5). Results suggest that GEMINI generally has better performance without sampling when generating critical subgraphs. We conjecture the reason is that removing the sampling makes the training of GEMINI more stable. Since the MI constraint $I_{GCLUB}$ is based on multiple samples of critical subgraphs, the Gumbel-Softmax sampling will result in a larger variance compared with directly using probabilities as weights. Moreover, Fig. 3(d) indicates that the no-sampling version of GEMINI achieves smaller edge and node probabilities than the counterpart with sampling and the GSAT baseline. The results indicate that $L_{sp}$ term is relatively more important in the no-sampling version of GEMINI because of more stable training as well, resulting in lower edge and node probabilities which are favorable to interpretability. Furthermore, smaller edge probabilities in Fig. 3(d) are also consistent with better interpretation AUCs in Fig. 3(c).

# 6 RELATED WORK

Methods endowing GNNs with interpretability could be roughly divided into two categories, i.e., post-hoc and built-in interpretability. Post-hoc methods (or explainers) analyze a fixed GNN model after training. These methods are generally implemented based on gradients (Sundararajan et al., 2017; Selvaraju et al., 2017) or perturbations (Ying et al., 2019; Luo et al., 2020) For example, GNNExplainer (Ying et al., 2019) generates a mask on the input graph based on an interpretation-oriented objective. However, post-hoc methods suffer from the faithfulness and instability w.r.t the model to be interpreted (Adebayo et al., 2021). Hence, GNNs with built-in interpretability are attracting rising attention. Wu et al. (2022) learns an invariant subgraph for interpretability based on invariant learning (IR). However, IR suffers from high computational costs and instability. Instead, a bunch of works resort to GIB for self-interpretability. The key is to estimate MI between graph variables. Yu et al. (2020; 2021) utilize the DV representation of KL divergence to estimate the MI between graphs, which results in heavy computational cost in practical training. Another way of bounding MI is to designate a prior distribution for critical subgraph either in graph space (Miao et al., 2022; Wu et al., 2020) or in embedding space (Sun et al., 2022), However, the introduced prior generally results in loose estimation of MI between graphs. Some works derive a closed-form of MI upper bound under strict restrictions (Yu et al., 2022; Lee et al., 2023) based on noise injection into node representations. However, the noise distribution should be designated as well and may be biased. Moreover, the graph structure information cannot be masked out effectively in these works. Briefly, existing works applying GIB suffer a lack of efficient estimation of upper bound MI between graph variables. Our work tackles this challenge through a combination of MI upper bound estimator and a variational distribution for graphs, which has largely been overlooked previously.

# 7 CONCLUSION

In this work, we have proposed a new framework of self-interpretable graph learning based on the information bottleneck principle. The key challenge of implementing the information bottleneck principle on graphs lies in calculating the mutual information efficiently and effectively for non-regular graph random variables. To overcome this obstacle, we develop a novel graph mutual information upper bound estimator. We extensively evaluated the performance of GEMINI and mainstream baselines on eight datasets with both interpretation and prediction metrics. Results reveal that the MI upper bound objective of GEMINI could improve the model's interpretability over graph sparsity constraints in most cases while maintaining comparable predictive capability.

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

# A  APPENDIX

## A.1  A DETAILED ANALYSIS ON THE CALCULATION OF $p_\phi(G_{\text{SUB}}|G)$

To rigorously illustrate the difficulty of calculating the exact values of $P(G_{\text{sub}}|G)$ and the necessity of the proposed variational distribution $q_\theta(G_{\text{sub}}|G)$, we need to introduce several definitions first. Note that we use upper case (e.g., $G$, $G_{\text{sub}}$) to represent graph random variables, and use lower case (e.g., $g_{\text{sub}}$, $g$, $g_1$, $g_2$) to indicate graph instances.

### A.1.1  SUBGRAPH MATCHING

Let's consider two graph instances, $g_1$ and $g_2$. The node sets of $g_1$ and $g_2$ are represented by $V(g_1)$ and $V(g_2)$ respectively, while their edge sets are denoted by $E(g_1)$ and $E(g_2)$. We assume that $|V(g_1)| \leq |V(g_2)|$. A valid subgraph matching $m$ from $g_1$ to $g_2$ is an injective function that maps the nodes of $g_1$ to nodes of $g_2$, such that for each edge $e_{u,v}$ in $E(g_1)$, the mapped edge $e_{m(u),m(v)}$ is still in $E(g_2)$. The set of all possible subgraph matching from $g_1$ to $g_2$ is defined as $M_{g_1 \to g_2}$, where

$$M_{g_1 \to g_2} \triangleq \{m | \text{for all } (u,v) \in V(g_1) \times V(g_1), e_{m(u),m(v)} \in E(g_2) \text{ if } e_{u,v} \in E(g_1), \}.$$

### A.1.2  RANDOM SUBGRAPH

Here we illustrate how to obtain an instance of the random subgraph $G_{\text{sub}}$ from $g$. Given a graph instance $g$, we assume that $p(v)$ represents the probability that the node $v$ is included in the subgraph, and $p(e_{u,v})$ as the probability that the edge $e_{u,v}$ is included. A common sampling procedure to obtain a subgraph instance $g_{\text{sub}}$ is:

1. Each node $v \in V(g)$ is sampled by a Bernoulli distribution $Bern(p(v))$.
2. For all sampled nodes in the first step, the edge $e_{u,v}$ is then sampled by the Bernoulli distribution $Bern(p(e_{u,v}))$. If there is no edge between $u$ and $v$ on $g$, $p(e_{u,v}) = 0$.

Let $p_\phi(G_{\text{sub}}|g)$ denote the probability mass function of the above random graph. The parameter $\phi$ in $p_\phi(G_{\text{sub}}|g)$ determines the values of $p(v)$ and $p(e_{u,v})$ in the sampling procedure. For example, $\phi$ is the parameter set of a neural network that outputs these probabilities.

For a graph instance $g$, we regard $\Omega_g$ as the sample space of $G_{\text{sub}}$ from the above random sampling procedure, satisfying that for any $g_{\text{sub}} \in \Omega_g$, $M_{g_{\text{sub}} \to g} \neq \emptyset$. Note that for a $g_{\text{sub}} \in \Omega_g$, $|V(g_{\text{sub}})|$ can be smaller than $|V(g)|$.

### A.1.3  THE EXACT CALCULATION OF $p_\phi(g_{\text{SUB}}|g)$

For a subgraph instance $g_{\text{sub}}$, we define

$$P_\phi^V(g_{\text{sub}}, g, m) = \prod_{v \in V(g_{\text{sub}})} p(m(v)) \prod_{v \in V(g)/V(g_{\text{sub}})} (1 - p(v))$$

to indicate the probability of nodes in $g_{\text{sub}}$ to be preserved from $g$ under the mapping $m$. Similarly, we define

$$P_\phi^E(g_{\text{sub}}, g, m) = \prod_{u,v \in V(g_{\text{sub}})} \mathbb{1}_{(u,v) \in E(g_{\text{sub}})} p(e_{m(u),m(v)}) + (1 - \mathbb{1}_{(u,v) \in E(g_{\text{sub}})})(1 - p(e_{m(u),m(v)}))$$

to indicate the probability of edges on $g_{\text{sub}}$ to be preserved from $g$ under the mapping $m$. Note that if $e_{m(u),m(v)}$ doesn't exist on $g$, we can regard the probability $p(e_{m(u),m(v)})$ as 0. Overall, for a subgraph instance $g_{\text{sub}} \in \Omega_g$, the probability mass function $p_\phi(g_{\text{sub}}|g)$ is calculated by

$$p_\phi(g_{\text{sub}}|g) = \sum_{m \in M_{g_{\text{sub}} \to g}} P_\phi^V(g_{\text{sub}}, g, m) \times P_\phi^E(g_{\text{sub}}, g, m)$$

The summation indicates that *all possible subgraph matchings* from $g_{\text{sub}}$ to $g$ should be considered when calculating the probability mass function $p_\phi(g_{\text{sub}}|g)$.

## A.2 THE LIMITATION OF EXISTING WORKS

The main challenge in computing $p_\phi(g_{\text{sub}}|g)$ lies in the $M_{g_{\text{sub}} \to g}$, which requires a subgraph matching algorithm to find all valid matchings. In previous works, $g_{\text{sub}}$ is always regarded as an edge-weighted variant of $g$, and the number of nodes in $g_{\text{sub}}$ is equivalent to that of $g$. In that case, the subgraph matching set $M_{g_{\text{sub}} \to g}$ degrades into a one-element set $\{\bar{m}\}$, where $\bar{m}$ is implicitly assumed to be known. Specifically, $\bar{m}$ is the mapping of each node of the weighted $g_{\text{sub}}$ to the corresponding node of the unweighted $g$. However, for the smaller subgraphs $g_{\text{sub}}$, where $|V(g_{\text{sub}})| < |V(g)|$, there may exist multiple valid subgraph matchings, i.e., $|M_{g_{\text{sub}} \to g}| > 1$. Considering only one mapping $\bar{m}$ leads to an inaccurate estimation of $p_\phi(g_{\text{sub}}|g)$.

Moreover, defining a prior distribution $r(G_{\text{sub}})$ over graphs is equally challenging as computing $p_\phi(g_{\text{sub}}|g)$. Designating a real prior graph distribution requires defining a graph generation process, and evaluating the likelihood of an instance under such a prior distribution encounters similar difficulties when calculating $p_\phi(g_{\text{sub}}|g)$. Hence, avoiding the graph prior is crucial for the graph MI estimation or calculating MI bounds.

## A.3 THE RATIONALE OF THE $q_\theta(g_{\text{SUB}}|g)$ FOR APPROXIMATING $p_\phi(g_{\text{SUB}}|g)$

Instead of calculating $p_\phi(g_{\text{sub}}|g)$ by finding the exact $M_{g_{\text{sub}} \to g}$, we approximate $p_\phi(g_{\text{sub}}|g)$ by a variational distribution $q_\theta(g_{\text{sub}}|g)$, which is implemented by graph neural networks. The reason is that the computation of $M_{g_{\text{sub}} \to g}$ is extremely time-consuming and intractable (NP-hard) for general graphs. In addition, we use the CLUB to calculate the upper bound of $I(G_{\text{sub}}; G)$, which solely relies on $q_\theta(g_{\text{sub}}|g)$ and avoids considering the prior $r(G_{\text{sub}})$.

## A.4 TRAINING ALGORITHM OF GEMINI

---

**Algorithm 1:** The training scheme of GEMINI

---

**Input** : Data distribution $p(G, Y)$, networks $g_\phi$, $q_\theta$, and $q_\omega$, number of training epochs $N_P$, number of batches $N_B$, batch size $K$.
**Output:** Optimized parameters $\phi$, $\theta$, $\omega$.
**for** $i = 1, \cdots, N_P$ **do**
    **for** $j = 1, \cdots, N_B$ **do**
        Sample a mini-batch of data $\{(G^{(1)}, Y^{(1)}), \cdots, (G^{(K)}, Y^{(K)})\}$ from $p(G, Y)$.
        Optimize $q_\theta$ by minimizing the objective in Eq. 8; fix $g_\phi$ (and $q_\omega$).
        Optimize $q_\omega$ by maximizing the objective in Eq. 4; fix $g_\phi$ (and $q_\theta$).
        Optimize $g_\phi$ by minimizing the objective in Eq. 13; fix $q_\theta$ and $q_\omega$.
    **end**
**end**

---

## A.5 DETAILS OF THE ADOPTED DATASETS.

We elaborate on the details of the adopted datasets in the experiments: (1) MUTAG (Debnath et al., 1991) is a molecule property prediction dataset. Each molecule graph is labeled as mutagenic or not. (2) MNIST-75sp (Knyazev et al., 2019) is an image classification dataset converted from images in the MNIST dataset. Each graph is with no more than 75 superpixels in which digital superpixels indicate ground truth explanations. (3) MOLHIV (Wu et al., 2018; Hu et al., 2020) is another molecule dataset for which the goal is to predict whether a molecule has the ability to inhibit HIV replication. (4) Graph-SST2 (Yuan et al., 2020) is a graph sentiment classification dataset in which each graph is converted from a sentence. Nodes represent tokens and edges indicate token correlations. (5) Spurious-Motif (Ying et al., 2019; Wu et al., 2022) s a synthetic dataset including specific motif patterns that determine graph labels. Each graph instance also includes a background pattern, which has a spurious correlation with the motif pattern. The spurious correlation strength is controlled by a parameter $b$ indicating that a motif and a specific background pattern exist simultaneously with probability $b$. We denote this dataset with parameter $b$ as *SPMotif (b)*. Three kinds of motifs (the *5-node house*, *5-node circle*, and *5-node crane*) and three background patterns (the *tree*, *wheel*, and *ladder*) are predetermined. The spurious relation does not exist in the test set.

Table 3: Comparison of spurious score between GEMINI and GSAT.

| | SPMotif (0.9) | | SPMotif (0.7) | |
|---|---|---|---|---|
| | Spurious Score | Prediction Score | Spurious Score | Prediction Score |
| GSAT | 0.0985 | 0.7517 | 0.0108 | 0.7711 |
| GEMINI | 0.0708 | 0.7776 | 0.0029 | 0.9123 |

## A.6 SPURIOUS SCORE COMPARISON

To investigate the resistance to spurious signals appearing in datasets, we calculate the *spurious score* and *predictive score* for the proposed method GEMINI and GSAT. We select three checkpoints from SPMotif (0.9) and SPMotif (0.7) for both methods and randomly generate 3000 spurious graphs for evaluation. Each spurious graph is generated by randomly selecting a motif graph and a base graph and then assembling the two. For each spurious graph $g_i$, we denote its label as $y_i$ (determined by its motif graph) and its spurious-aligned class as $s_i$ (determined by its base graph). The spurious-aligned class is the label of motif graph with which a spurious base graph is correlated in the training set SPMotif (0.9) and SPMotif (0.7). For example, in SPMotif (0.9), a spurious base graph *tree* has a probability of 0.9 to appear with a motif graph *house* together. Since the label for *house* is 0, the spurious-aligned class for *tree* is 0. In the evaluation dataset, we ensure $y_i \neq s_i$ for each $g_i$. Referring to the established work (Adebayo et al., 2021), we define the **spurious score** as follows:

$$SpurScore = \frac{1}{N} \sum_{i=1}^{N} f(g_i)[s_i] \tag{14}$$

where $f(g_i)$ is the output probabilities of a model, i.e., a three-dimensional vector in our setting. $N$ is the number of evaluation samples, i.e., 3000 in this experiment. The spurious score indicates the average probability of a model mistakenly predicting a graph to its spurious class. Similarly, we calculate the **prediction score** as follows.

$$PredScore = \frac{1}{N} \sum_{i=1}^{N} f(g_i)[y_i] \tag{15}$$

The prediction score indicates the average probability of a model correctly predicting a graph to its groundtruth class.

The results are shown in Table 3, we can find that for models trained from both datasets, the spurious score of GEMINI is smaller than that of GSAT. Moreover, the prediction score of GEMINI is significantly better than that of GSAT, especially when the spurious signal is moderate, e.g., SPMotif (0.7). These results verify the superior resistance and robustness against spurious signals compared to GSAT. In the future, we hope to theoretically analyze the capability and possibility of the proposed methods regarding the removal of spurious information.

