# OpenReview forum: "Enhancing Mutual Information Estimation in Self-Interpretable Graph Neural Networks"
_ICLR.cc/2024/Conference — Submitted to ICLR 2024_

### Official Review · Reviewer_BVpy · 2023-10-30

**Soundness:** 3 good
**Presentation:** 2 fair
**Contribution:** 2 fair
**Rating:** 5
**Confidence:** 4

**Summary:**

Existing self-interpretable Graph Neural Networks (GNNs) built upon Graph Information Bottleneck (GIB) suffer from the burdensome of mutual information estimation. To address this issue, this work proposes a novel framework for self-interpretable GNNs with an enhanced technique for mutual information estimation, namely GENIMI. Experiment results indicate the proposed GENIMI enjoys improved predictive and interpretable performance.

**Strengths:**

This paper is well-written and easy to follow. The motivation for improving the mutual information estimation in the GIB framework is clear and crucial. Empirical results show that the proposed GEMINI enjoys competitive performances of GNN prediction and interpretability.

**Weaknesses:**

However, the reviewer is concerned with some theoretical details.

1. For the predictive term in Eqn. 3, the appropriate formula derivation is: $I(G_{sub};Y)=E_{p(G_{sub},Y)}\log{\frac{p(Y|G_{sub})}{p(Y)}}\geq E_{p(G_{sub},Y)}q_{\omega}(Y|G_{sub})+H(Y)$.

2. Does $p_{\phi}(G_{sub}|G)$ and $q_{\theta}(G_{sub}|G)$ share the same subgraph generator? If so, what is the intuition behind using $q_{\theta}(G_{sub}|G)$ to approach $p_{\phi}(G_{sub}|G)$?

**Questions:**

The authors are encouraged to address the concerns in Weaknesses.

---

> ### Author Response · Authors · 2023-11-17
> **Response to reviewer BVpy**
>
> >Q1:For the predictive term in Eq. 3, the appropriate formula derivation is: $I(G_{sub};Y) = E_{p(G_{sub},Y)} \log \frac{p(Y|G_{sub})}{p(Y)} \geq E_{p(G_{sub},Y)} \log q_\omega(Y|G_{sub}) + H(Y)$.
>
> A1: Thanks for your comments. We apologize for the error in Eqn. 3, where the denominator of the first term of line 1 should be $p(Y)$ instead of $p(G_{sub})$. We have revised it in the paper. Please note, however, that this mistake does not impact the following derivations and conclusions presented in the paper.
>
>
> >Q2: Does $p_\phi(G_{sub}|G)$ and $q_\theta(G_{sub}|G)$ share the same subgraph generator? If so, what is the intuition behind using $q_\theta(G_{sub}|G)$ to approach $p_\phi(G_{sub}|G)$?
>
> A2: Thanks for your question! $p_\phi(G_{sub}|G)$ is the probability that $G_{sub}$ is sampled by $G$ based on a underlying subgraph generation process parameterized by $\phi$. However, learning $\phi$ is not enough to calculate the exact value of $p_\phi(G_{sub}|G)$, it should also include a subgraph matching step to compute $p_\phi(G_{sub}|G)$, leading to a NP-hard complexity. Previous works (e.g., GSAT) made some assumptions to ignore the subgraph matching step. Nonetheless, this has resulted in an imprecise learning of $\phi$ and an inaccurate calculation of $I(G_{sub};G)$.
>
> Instead, we simultaneously learn $\phi$ and approximate $p_\phi$ by a variational distribution $q_\theta$. Based on Eq.7 in the paper, the maximization of $E_{p(G, G_{\text{sub}})} [ \log q_\theta( G_{\text{sub}}|G) ]$ is equivalent to minimizing $E_{p(G)}[KL(p_\phi(G_{\text{sub}}|G) || q_\theta(G_{\text{sub}}|G))]$. Thus, we learn $q_\theta$ in Eq.8 with a fixed $\phi$. The optimization of $\phi$ is achieved by minimizing the total GIB objective.
> Please refer to the ***Common concern 1*** for a rigorous analysis.
>
>
> In the detailed implementation, the subgraph generator $g_\phi$ is used to instantiate the subgraph generation process, and hence determines the distribution $p_\phi(G_{sub}|G)$. $q_\theta$ is implemented as a neural network taking $G_{sub}$ and $G$ as inputs and outputs a probability. The network $q_\theta$ includes two GNNs to extract representations of $G_{sub}$ and $G$ respectively and then calcutate a probability. The parameters in $g_\phi$ and $q_\theta$ are learned iteratively, as depicted in the pseudo-code in Appendix.

---

### Official Review · Reviewer_NeoE · 2023-10-30

**Soundness:** 3 good
**Presentation:** 3 good
**Contribution:** 3 good
**Rating:** 6
**Confidence:** 2

**Summary:**

The goal of the paper is to evaluate the mutual information (MI) between an input graph and a key subgraph. To tackle this problem, the authors propose a novel framework called GEMINI, which trains self-interpretable graph models and addresses the challenge of distorted and imprecise estimations in graph MI estimation research. The authors construct a variational distribution over the critical subgraph and create an effective MI upper bound estimator. The proposed method is shown to be effective according to empirical results.

**Strengths:**

1. This paper is well-organized and well-written. The authors provide sufficient details about their work and easy to understand.

2. Estimating the mutual information between the input graph and the subgraph is both important and challenging.

**Weaknesses:**

1. This work closely follows GSAT[1]. Its main theoretical contribution is the addition of the information bottleneck (IB) upper bound loss $L_{GCLUB}$ to the objective of GSAT[1], which is based on the idea of variational CLUB[2].

2. Does the proposed model's ability to remove the spurious correlation come from the framework of GSAT? Can GEMINI provide a theoretical guarantee for the removal of spurious correlations?

3. Some of the numerical results reported in Table 1 and Table 2 are quite different from those reported in GSAT. The differences are particularly noticeable with the numbers that involve MNIST-75sp in Table 1 and those associated with SPMotif in Table 2, relating to GIN and GSAT. It would be helpful if the authors could provide further details about their implementations and explanations for these differences.

[1] Miao, Siqi, Mia Liu, and Pan Li. "Interpretable and generalizable graph learning via stochastic attention mechanism." In International Conference on Machine Learning, pp. 15524-15543. PMLR, 2022.
[2] Cheng, Pengyu, Weituo Hao, Shuyang Dai, Jiachang Liu, Zhe Gan, and Lawrence Carin. "Club: A contrastive log-ratio upper bound of mutual information." In International conference on machine learning, pp. 1779-1788. PMLR, 2020.

**Questions:**

1. On page 3, in the last sentence before Eq.3, should it be a “lower” bound of $I(G_{sub};Y)$?

2. I cannot find the curve of GSAT in the second subfigure of Fig. 2(d). Is it missing or unavailable?

---

> ### Author Response · Authors · 2023-11-17
> **Response to reviewer NeoE (part 1)**
>
> >Q1: This work closely follows GSAT[1]. Its main theoretical contribution is the addition of the information bottleneck (IB) upper bound loss $I_{GCLUB}$ to the objective of GSAT[1], which is based on the idea of variational CLUB[2].
>
> A1: Thanks for your comments. The novelty of the paper is that we point out the limitation of GSAT and propose a framework to approximate $p_\phi(G_{sub}|G)$ by the variational distribution $q_\theta(G_{sub}|G)$.
>
>
> In detail, GSAT assumed the generated subgraph $G_{sub}$ is an edge-weighted variant of $G$, where the nodes in $G_{sub}$ and $G$ are one-to-one correspondence, and the edges in $G$ are sampled to obtain $G_{sub}$. However, these works overlook the scenario where the number of nodes in $G_{sub}$ is less than that in $G$. Even if the number of nodes in $G_{sub}$ equates to that of $G$, the correspondence between nodes in $G_{sub}$ and those in $G$ may be nonunique. In such cases, it is necessary to perform subgraph matching before calculating the probability $p_\phi(G_{sub}|G)$. Since subgraph matching is NP-hard, it becomes extremely difficult to accurately compute the value of $p_\phi(G_{sub}|G)$. Given the learned $\phi$, previous studies have not been able to compute an effective value of $p_\phi(G_{sub}|G)$.
> Moreover, computing the prior distribution $r(G_{sub})$ over graphs is equally challenging as $p_\phi(G_{sub}|G)$. GSAT involves the term $r(G_{sub})$ to compute the upper bound value of $I(G_{sub};G)$. However, the calculation of $r(G_{sub})$ requires to designate a random graph sampling process and hence encounters similar difficulties as when calculating $p_\phi(G_{sub}|G)$.
>
> Instead, our proposed method approximates $p_\phi$ by a variational distribution $q_\theta$. Based on Eq.7 in the paper, the maximization of $E_{p(G, G_{\text{sub}})} \left[ \log q_\theta( G_{\text{sub}}|G) \right]$ is equivalent to minimizing $E_{p(G)}[KL(p_\phi(G_{\text{sub}}|G)$ || $q_\theta(G_{\text{sub}}|G))]$. Thus, we learn $q_\theta$ in Eq.8. In addition, we use the CLUB to calculate the upper bound of $I(G_\text{sub};G)$, which avoids considering the prior $r(G_\text{sub})$.

---

> ### Author Response · Authors · 2023-11-17
> **Response to reviewer NeoE (part 2)**
>
> >Q2: Does the proposed model's ability to remove the spurious correlation come from the framework of GSAT? Can GEMINI provide a theoretical guarantee for the removal of spurious correlations?
>
> A2: Theoretically justifying the capability of the proposed method for removing spurious signals can be challenging, since we need to mathematically formalize the *removal*  of spurious signals. Therefore, we conducted an experiment to empirically compare the resistance to spurious signals between the proposed method and GSAT. We have also added a section in Appendix A.6.
>
> Specifically, we select three checkpoints from SPMotif (0.9) and SPMotif (0.7) for both methods and randomly generate 3000 spurious graphs to evaluate the **spurious score** and the **predictive score**. Each spurious graph is generated by randomly selecting a motif graph and a base graph and then assembling the two. For each spurious graph $g_i$, we denote its label as $y_i$ (determined by its motif graph) and its spurious-aligned class as $s_i$ (determined by its base graph). The spurious-aligned class is the label of motif graph with which a spurious base graph is correlated in the training set SPMotif (0.9) and SPMotif (0.7). For example, in SPMotif (0.9), a spurious base graph *tree* has a probability of 0.9 to appear with a motif graph *house* together. Since the label for *house* is 0, the spurious-aligned class for *tree* is 0.
>
> In the evaluation dataset, we ensure $y_i \neq s_i$ for each $g_i$. Referring to the established work (Post hoc explainers may be ineffective for detecting unknown spurious correlation), we define the **spurious score** as
>
> $
> SpurScore = \frac{1}{N}\sum_{i=1}^{N} f(g_i)[s_i].
> $
>
> where $f(g_i)$ is the output probabilities of a model, i.e., a three-dimensional vector in our setting. $N$ is the number of evaluation samples, i.e., 3000 in this experiment. The spurious score indicates the average probability of a model mistakenly predicting a graph to its spurious class.
> Similarly, we calculate the **predictive score** as
>
> $
> PredScore = \frac{1}{N}\sum_{i=1}^{N} f(g_i)[y_i].
> $
>
> The predictive score indicates the average probability of a model correctly predicting a graph to its ground truth class.
>
> The results are shown in the table:
> |        |  SPMotif (0.9) |                  |  SPMotif (0.7) |                  |
> |:------:|:--------------:|:----------------:|:--------------:|:----------------:|
> |        | Spurious Score | Predictive Score | Spurious Score | Predictive Score |
> |  GSAT  |     0.0985     |      0.7517      |     0.0108     |      0.7711      |
> | GEMINI |     0.0708     |      0.7776      |     0.0029     |      0.9123      |
>
> We find that for models trained from both datasets, the spurious score of GEMINI is smaller than that of GSAT. Moreover, the predictive score of GEMINI is significantly better than that of GSAT, especially when the spurious signal is moderate, e.g., SPMotif (0.7). These results verify the superior resistance and robustness against spurious signals compared to GSAT.

---

> ### Author Response · Authors · 2023-11-17
> **Response to reviewer NeoE (part 3)**
>
> >Q3: Some of the numerical results reported in Table 1 and Table 2 are quite different from those reported in GSAT. The differences are particularly noticeable with the numbers that involve MNIST-75sp in Table 1 and those associated with SPMotif in Table 2, relating to GIN and GSAT.
>
> A3: Thanks for your comments! For the SPMotif results in Table 2, there is a parameter to control the size of the base (spurious) graph in the generation of SPMotif. In GSAT, the size is relatively larger, e.g., 20 for tree, 60 for wheel, in the test set. We find that a large base graphs prone to result in distribution shift. We slightly adjust the parameter to be smaller, e.g., 15 for tree, 30 for wheel, to relieve the distribution shift and facilitate comparison. Because we mainly aim to focus on the spurious correlation and interpretation performance, instead of distribution shifts. Moreover, we generate more samples for SPMotif compared with the implementation of GSAT. We generate 30000 samples for a SPMotif dataset, while the number is 3000 in GSAT according to their source code. Hence, the predictive performance of GSAT on SMPotifs in Table 2 is better than their originally reported result.
>
> For the MNIST-75sp results in Table 1, We adopt the strictly identical dataset settings as with the GSAT baseline. The difference may come from some model parameters, adopted loss coefficients, and randomness, since the dataset size of MNIST-75sp is only 20000 and we only run three times to obtain the averaged results, which may also affect the evaluation numbers.
>
>
> >Q4: On page 3, in the last sentence before Eq.3, should it be a “lower” bound of $I(G_{sub};Y)$?
>
> A4: Yes, you are correct. We are sorry for this typo and have made a revision.
>
> >Q5: I cannot find the curve of GSAT in the second subfigure of Fig. 2(d). Is it missing or unavailable?
>
> A5: The second subfigure Fig. 2(d) compares node probabilities. GSAT solely learns edge probabilities, and hence node probabilities remain 1.0. The node probability for GIN is also 1.0. So the curve of GIN  overlaps over the curve of GSAT. We have added a clarification in the caption of Fig. 2(d).

---

### Official Review · Reviewer_51zY · 2023-10-31

**Soundness:** 3 good
**Presentation:** 2 fair
**Contribution:** 3 good
**Rating:** 6
**Confidence:** 3

**Summary:**

The Graph Information Bottleneck framework significantly enhances the self-interpretability of Graph Neural Networks. However, current approaches in estimating the mutual information between graph explanations and their original forms frequently yield distorted and imprecise estimations, ultimately compromising the effectiveness of the model. In response to these limitations, this paper introduces a novel framework called GEMINI to address these challenges.

**Strengths:**

+ They utilize a MI upper bound estimator based exclusively on the conditional probability distribution.
+ They introduce a variational distribution and its suitable instantiation for the conditional probability distribution.
+ Extensive experiments demonstrate the effectiveness of the proposed framework.

**Weaknesses:**

-	They employ established MI estimator theory, which appears easily extendable to the graph domain. In my view, it seems they have not drawn particularly interesting conclusions or specific designs for graphs. I have some reservations about the novelty of the proposed framework.
-	The experimental results are not convincing enough. SOTA explainers for GNNs should be set as baselines. Moreover, the proposed model did not exhibit a significant improvement compared to these baselines.
-	The paper's writing and organization require enhancements. For instance, it is challenging for readers to discern the corresponding relationships between the limitations and the contributions.

**Questions:**

Please refer to the weaknesses.

- They employ established MI estimator theory, which appears easily extendable to the graph domain. In my view, it seems they have not drawn particularly interesting conclusions or specific designs for graphs. I have some reservations about the novelty of the proposed framework.
- The experimental results are not convincing enough. SOTA explainers for GNNs should be set as baselines. Moreover, the proposed model did not exhibit a significant improvement compared to these baselines.
- The paper's writing and organization require enhancements. For instance, it is challenging for readers to discern the corresponding relationships between the limitations and the contributions.

---

> ### Author Response · Authors · 2023-11-17
> **Response to reviewer 51zY**
>
> >Q1: They employ established MI estimator theory, which appears easily extendable to the graph domain. In my view, it seems they have not drawn particularly interesting conclusions or specific designs for graphs. I have some reservations about the novelty of the proposed framework.
>
>
> A1: Thanks for your comments. The novelty of the paper is that we point out the limitation of previous GIB works and propose a framework to approximate $p_\phi(G_{sub}|G)$ by the variational distribution $q_\theta(G_{sub}|G)$. The experimental results demonstrate that we can have a better estimation of the GIB term compared with GSAT and DIR.
>
> In detail, previous GIB works think the generated subgraph $G_{sub}$ is an edge-weighted variant of $G$, where the nodes in $G_{sub}$ and $G$ are one-to-one correspondence, and the edges in $G$ are sampled to obtain $G_{sub}$. However, these works overlook the scenario where the number of nodes in $G_{sub}$ is less than that in $G$. Even if the number of nodes in $G_{sub}$ equates to that of $G$, the correspondence between nodes in $G_{sub}$ and those in $G$ may be nonunique. In such cases, it is necessary to perform subgraph matching before calculating the probability $p_\phi(G_{sub}|G)$. Since subgraph matching is NP-hard, it becomes extremely difficult to accurately compute the value of $p_\phi(G_{sub}|G)$. Given the learned $\phi$, previous studies have not been able to compute an effective value of $p_\phi(G_{sub}|G)$, which affects the upper bound value of $I(G_{sub};G)$. Instead, our proposed method approximates $p_\phi$ by a variational distribution $q_\theta$ with the learned $\phi$. Based on Eq.7 in the paper, the maximization of $E_{p(G, G_{\text{sub}})} \left[ \log q_\theta( G_{\text{sub}}|G) \right]$ is equivalent to minimizing $E_{p(G)}[KL(p_\phi(G_{\text{sub}}|G)$ || $q_\theta(G_{\text{sub}}|G))]$. Thus, we learn $q_\theta$ in Eq.8.
>
> For a formal analysis of the limitations of previous works, please refer to ***Common concern 1***.
>
> >Q2: The experimental results are not convincing enough. SOTA explainers for GNNs should be set as baselines. Moreover, the proposed model did not exhibit a significant improvement compared to these baselines.
>
>
> A2: Thank you for your feedback. Indeed, GSAT is the state-of-the-art baseline with both predictive and interpretable capabilities.
> Additionally, GNNExplainer and PGExplainer are frequently employed as post-hoc explanation methods for GNNs in prior research. As demonstrated in Table 1 & 2, our proposed method maintains comparable predictive performance with mainstream GNNs (e.g., GIN), and improves the interpretation AUC over the second-best baseline across five datasets. The average improvement is about 3%$\sim$ 5% over the state-of-the-art self-interpretable baseline. Thus, we believe our model exhibits a significant advancement in interpretability, contributing to the existing body of research on GNNs.
>
> >Q3: The paper's writing and organization require enhancements. For instance, it is challenging for readers to discern the corresponding relationships between the limitations and the contributions.
>
>
> A3: Thanks for your suggestions. We have revised the paper more clearly to distinguish the limitation analysis and our proposed method. We also added the rigorous formulation and limitation analysis in Appendix A.1. Please refer to Section 3, 4, and Appendix A.1 in the revised paper.

---

### Official Review · Reviewer_6KN9 · 2023-10-31

**Soundness:** 2 fair
**Presentation:** 3 good
**Contribution:** 2 fair
**Rating:** 5
**Confidence:** 3

**Summary:**

In this paper, the authors introduce a novel approach for approximating the Graph Information Bottleneck (GIB). Their method focuses on modeling the distribution of arbitrary subgraphs and graphs, while also bypassing the need to model the prior of subgraphs. Experimental results seem to demonstrate the effectiveness of the proposed method.

**Strengths:**

1. The paper is well-written, with the authors providing a thorough derivation of the proposed GIB approximation and presenting a clear step-by-step explanation of their method.

2. The authors employ the CLUB technique to circumvent the need for modeling the prior of subgraphs. This approach appears to relax the assumptions made in previous methods, enhancing the flexibility and applicability of the proposed approach.

**Weaknesses:**

1. The model architecture in this paper bears a resemblance to GSAT, and it would be advantageous if the authors could explicitly delineate the key distinctions between the two. Furthermore, the experimental results suggest a notable enhancement over GSAT despite their similar model architectures. Providing inference codes for model reproduction would greatly facilitate the validation of these results and contribute to the paper's overall reproducibility and transparency.

2. The authors assert that the proposed method can generate sparse subgraphs even without the need for sparse regularization. However, it is evident that L_sp is introduced as a subgraph term to regulate graph sparsity. In the ablation study, the authors argue that this term is essential, which appears to be inconsistent with their initial claim in the introduction.

**Questions:**

1. See the comments above.

2. What is the benefit of modeling arbitrary subgraphs and graphs? Since G_{sub}^1 should be sampled from G_1 and should not be related to G_2.

3. Why is the MI upper bound approximation proposed in the method better than previous methods?

---

> ### Author Response · Authors · 2023-11-17
> **Response to reviewer 6KN9**
>
> >Q1: The model architecture in this paper bears a resemblance to GSAT, and it would be advantageous if the authors could explicitly delineate the key distinctions between the two. Furthermore, the experimental results suggest a notable enhancement over GSAT despite their similar model architectures. Providing inference codes for model reproduction would greatly facilitate the validation of these results and contribute to the paper's overall reproducibility and transparency.
>
> A1: Thanks for your advice! The key difference between GSAT and our proposed method is that we improve the calculation of $p_\phi(G_{sub}|G)$ and $I(G; G_{sub})$. Actually, $p_\phi(G_{sub}|G)$ is the probability that $G_{sub}$ is a generated subgraph from $G$ based on a subgraph generation process parameterized by $\phi$. However, knowing $\phi$ is not enough to calculate the exact value of $p_\phi(G_{sub}|G)$, it should also include a subgraph matching step to compute $p_\phi(G_{sub}|G)$, leading to a NP-hard complexity. GSAT made some assumptions to ignore the subgraph matching step. Nonetheless, this has resulted in an imprecise learning of $\phi$ and an inaccurate calculation of $I(G_{sub};G)$.
>
> Instead, we simultaneously learn $\phi$ and approximate $p_\phi$ by a variational distribution $q_\theta$. Based on Eq.7 in the paper, optimizing the objective in Eq. 8 minimizes the KL divergence between $p_\phi$ and $q_\theta$. The optimization of $\phi$ is achieved by minimizing the total GIB objective.
>
>
> Please refer to the ***Common concern 1*** for a comprehensive and rigorous analysis. For the inference codes, we have provided the source code in the supplementary material to facilitate the reproduction of our experimental results.
>
> >Q2: The authors assert that the proposed method can generate sparse subgraphs even without the need for sparse regularization. However, it is evident that L_sp is introduced as a subgraph term to regulate graph sparsity. In the ablation study, the authors argue that this term is essential, which appears to be inconsistent with their initial claim in the introduction.
>
> A2: Thanks for your comments. In the first section of the ablation study (Effect of the $L_{GCLUB}$ term), we remove the $L_{sp}$ term and rely solely on the $L_{GCLUB}$ regularization. Figure 2(b) demonstrates that $L_{GCLUB}$ could generate sparse graphs, e.g., the average edge/node probability is about $0.6\sim 0.7$. We claim $L_{sp}$ is essential for generating even ***sparser*** graphs, e.g., average edge/node probability of $0.3\sim 0.5$. The term $L_{sp}$ is used to provide users with the opportunity to achieve a desired level of subgraph sparsity based on their preferences.
>
> >Q3: What is the benefit of modeling arbitrary subgraphs and graphs? Since $G_{sub}^1$ should be sampled from $G_1$ and should not be related to $G_2$.
>
> A3: Thanks for your questions! Different from the previous works, we provide a more accurate calculation of $p_\phi(G_{sub}|G)$ and $I(G;G_{sub})$. In detail, we approximate $p_\phi(G_{sub}|G)$ by a variational distribution $q_\theta(G_{sub}|G)$, and use the CLUB upper bound for $I(G;G_{sub})$.
>
> When adopting our $I_{GCLUB}$ bound to calculate the upper bound of $I(G;G_{sub})$, a subgraph instance of $G_{sub}$ may not be directly sampled from a graph instance of $G$. According to the CLUB bound in Eq. 5 of the paper, for the second term $E_{p(G)} E_{p(G_{sub}) } [ \log p_\phi( G_{\text{sub}}|G )]$, $G_{sub}$ and $G$ come from their own marginal distributions. Hence when we calculate this term, we need to calculate $p_\phi( g_{\text{sub}}^2 |g^1 )$ where $g^1$ is sampled from $p(G)$, and $g_{\text{sub}}^2$ is sampled from $p(G_{\text{sub}})$. $g_{\text{sub}}^2$ may not come from $g^1$.
>
> >Q4: Why is the MI upper bound approximation proposed in the method better than previous methods?
>
> A4: Thanks for your questions! The reason for the utilization of CLUB bound is to avoid the estimation of the prior distribution $r(G_{sub})$. Computing $r(G_\text{sub})$ over graphs is equally challenging as $p_\phi(G_{sub}|G)$. For example, GSAT regarded the prior probability of a subgraph instance $r(g_{sub})$ as the product of edge probabilities on $g$, where the existence of each edge follows an independent Bernoulli distribution. However, $r(g_{sub})$ is essentially $r(g_{sub}|g)$, which relies on the parent graph $g$ (as we have stated, in GSAT and other works, $g_{sub}$ must be a weighted version of $g$ and the matching between node sets of $g_{sub}$ and $g$ is implicitly assumed to be known), and is not a *real* prior distribution over graphs. A real graph prior $r(g_{sub})$ should be able to be calculated for an arbitrary graph $g_{sub}$ without knowing its *parent graph* $g$. Hence, we should utilize an MI bound that does not depend on  $r(G_{sub})$.

---

### Author Response · Authors · 2023-11-17
**To all reviewers**

# Common concern 2: Core contributions
### 2.1 Point out limitations of existing methods for graph MI estimation

The key difficulty of exactly calculating $p_\phi(g_{sub}|g)$ relies on finding all subgraph matchings $M_{g_{sub}\rightarrow g}$. In previous works, $g_{sub}$ is always regarded as an edge-weighted version of $g$ and the number of nodes in $g_{sub}$ equates to that of $g$. In that case, the subgraph matching set $M_{g_{sub}\rightarrow g}$ actually degrades into a one-element set $\{\bar{m}\}$, where $\bar{m}$ is implicitly assumed to be known, i.e., the one that maps each node of the weighted $g_{sub}$ to the corresponding node of the unweighted $g$.  However, this is not true for general subgraphs, which may be smaller than $g$, i.e., $|V(g_{sub})|< |V(g)|$. Because for these smaller subgraphs, there may exist multiple subgraph matchings, i.e., $|M_{g_{sub}\rightarrow g}|>1$. Besides, even if the size of $g_{sub}$ equates to the size of $g$, the subgraph matching can also be non-unique.

Moreover, computing the prior distribution $r(G_\text{sub})$ over graphs is equally challenging as $p_\phi(G_{sub}|G)$. The previous works also involve the term $r(G_\text{sub})$ to compute the bounds of graph MI. For example, GSAT regarded the prior probability of a subgraph instance $r(g_{sub})$ as the product of edge probabilities on $g$, where the existence of each edge follows an independent Bernoulli distribution. However, $r(g_{sub})$ is essentially $r(g_{sub}|g)$, which relies on the parent graph $g$ (as we have stated, in GSAT and other works, $g_{sub}$ must be a weighted version of $g$ and the matching between node sets of $g_{sub}$ and $g$ is implicitly assumed to be known), and is not a *real* prior distribution over graphs. A real graph prior $r(g_{sub})$ should be able to be calculated for an arbitrary graph $g_{sub}$ without knowing its parent graph $g$. Hence, we should utilize an MI bound that does not depend on the calculation of $r(g_{sub})$.

### 2.2 Adopt $q_\theta(g_{sub}|g)$ to approximate $p_\phi(g_{sub}|g)$


In light of the above analysis, we have to either resort to a subgraph matching algorithm to find the exact $M_{g_{sub}\rightarrow g}$, or try to approximate the ground truth probability mass function $p_\phi(g_{sub}|g)$. Since calculating $M_{g_{sub}\rightarrow g}$ is extremely time-consuming and intractable (NP-hard) for general graphs, we adopt the later scheme, i.e., utilizing a variational distribution $q_\theta(g_{sub}|g)$ (which is implemented by a neural network) to approximate $p_\phi(g_{sub}|g)$.


Moreover, once $q_\theta(g_{sub}|g)$ is obtained, we need to calculate the upper bound of graph mutual information $I(G; G_{sub})$. Previous works require designating a prior $r(G_{sub})$ to obtain a graph MI upper bound, while $r(G_{sub})$ is also difficult to design over the graph domain. We adopt an MI upper bound (CLUB) which solely relies on $q_\theta(g_{sub}|g)$ to circumvent $r(G_{sub})$.

# Common concern 3: relation with GSAT

Our proposed method GEMINI indeed resembles GSAT in its computational architecture. However, we enhance the estimation of the upper bound of $I(G; G_{sub})$ in the graph information bottleneck framework. Specifically,

- GSAT calculated the probability $p_\phi(G_{sub}|G)$ by multiplying probabilities of the preserved edges in the subgraph without considering subgraph matching. It is problematic as we have analyzed above. Unlike GSAT, we adopt a variational $q_\theta(G_{sub}|G)$ to approximate this conditional probability and develop a learning objective to optimize $q_\theta$.

- GSAT utilized a prior distribution $r(G_{sub})$, which is hard to estimate. In our work, we utilize CLUB upper bound that does not depend on $r(G_{sub})$, circumventing the difficulty of designating a prior distribution over subgraphs.

---

### Author Response · Authors · 2023-11-17
**To all reviewers**

# Summary of the revision
We sincerely thank all the reviewers for the valuable comments, which help to improve the quality of our work. We summarize the revision in the updated version as follows:
- We revised section 4 to correct some typos and remove limitation analysis to a new section.
- We added the new section 3 to distinguish the limitation analysis of current graph MI estimations and our proposed method.
- We added a section in the Appendix to rigorously formulate the analysis in section 3.
- We revised the introduction to be consistent with our revised paper.
- We added an experiment to investigate the spurious score of different models.

# Common concern 1: Limitations of existing methods when calculating $p_\phi(G_{sub}|G)$

To rigorously illustrate the difficulty of calculating the exact values of $p_\phi(G_{sub}|G)$ and the necessity of the proposed variational distribution $q_\theta(G_{sub}|G)$, we need to introduce several definitions first. Note that we use upper case (e.g., $G$, $G_{sub}$) to represent graph random variables, and use lower case (e.g., $g_{sub}$, $g$, $g_1$, $g_2$) to indicate graph instances.

### 1.1 Subgraph matching
Let's consider two graph instances, $g_1$ and $g_2$. The node sets of $g_1$ and $g_2$ are represented by $V(g_1)$ and $V(g_2)$ respectively, while their edge sets are denoted by $E(g_1)$ and $E(g_2)$. We assume that $|V(g_1)| \leq |V(g_2)|$. A valid subgraph matching $m$ from $g_1$ to $g_2$ is an injective function that maps the nodes of $g_1$ to nodes of $g_2$, such that for each edge $e_{u,v}$ in $E(g_1)$, the mapped edge $e_{m(u),m(v)}$ is still in $E(g_2)$. The set of all possible subgraph matching from $g_1$ to $g_2$ is defined as $M_{g_1 \rightarrow g_2}$, where

$
M_{g_1\rightarrow g_2} \triangleq \{m| \text{for all } (u,v) \in V(g_1) \times V(g_1), e_{m(u),m(v)} \in E(g_2) \text{ if } e_{u,v} \in E(g_1) \}.
$


### 1.2 The probability distribution over subgraphs

For a graph instance $g$, assuming $p(e_{u,v})$ is the learned edge probability for $e_{u,v}\in E(g)$ and $p(v)$ is the learned node probability for node $v\in V(g)$. We consider the following procedure to obtain a subgraph instance $g_{sub}$ from $g$:
1. Sampling each node $v\in V(g)$ basd on a Bernoulli distribution $Bern(p(v))$.
2. For all sampled nodes in the first step, sampling edges $e_{u,v}$ between $u$ and $v$ based on a Bernoulli distribution $Bern(p(e_{u,v}))$. If there is no edge between $u$ and $v$ on $g$, the $p(e_{u,v})$ is regarded as 0.


In the following, we formally derivate the probability mass function $p_\phi(g_{sub}|g)$ on the subgraph sample space $\Omega_g$ induced by $g$ and edge/node probabilities (which is determined by $\phi$), and the probability space of random subgraphs sampled from $g$. Note that we mainly focus on the graph topology since the sampling procedure and learned probabilities only involve the graph substructure information.

For a graph instance $g$ with $n$ nodes, we denote the  subgraph space of $g$ as $\Omega_g$, which contains all possible subgraphs from $g$, i.e., $\Omega_{g} = \cup_{i=1}^{n} \Omega^i_g$ where $\Omega^i_g$ is the set of all possible non-induced subgraphs of $g$ with exactly $i$ nodes. In other words, for any $g_{sub} \in \Omega_g$, $M_{g_{sub}\rightarrow g} \neq \emptyset$.



Based on the sampling (or graph generation) process above, $g$ determines a **probability space** of a random subgraph variable, for which the sample space is $\Omega_g$, and the probability measure is implicitly determined. In the following, we formalize the probability mass function.


For a specific subgraph matching instance $m$, we define

$
P^{V}(g_{sub}, g, m) = \prod_{v\in V(g_{sub})} p(m(v)) \prod_{v\in V(g_{sub})/ V(g_{sub})} (1-p(v))
$

to indicate the likelihood of nodes in an instance $g_{sub}$ are preserved from $g$ under the mapping $m$, where $p(m(v))$ is the learned node probability for node $m(v)$, which belongs to $g$.

Similarly, we define

$
P^{E}(g_{sub}, g, m) = \prod_{u,v \in V(g_{sub})} 1_{(u, v)\in E(g_{sub})} p(e_{m(u), m(v)}) + (1-1_{(u, v)\in E(g_{sub})})  (1-p(e_{m(u), m(v)}))
$

to indicate the probability of edges on the instance $g_{sub}$ to be preserved from $g$ under the mapping $m$, where $p(e_{m(u), m(v)})$ is the learned edge probability for edge $e_{m(u), m(v)}$, which belongs to $g$. The $1_{condition}$ is the indicator function.
Note that if $e_{m(u), m(v)}$ doesn't exist on $g$, we can regard the probability $p(e_{m(u), m(v)})$ as 0.

The probability mass function $p_\phi(g_{sub}|g)$ on $\Omega_{g}$ should be

$
p_\phi(g_{sub}|g) = \sum_{m \in M_{g_{sub}\rightarrow g}}
P^{V}(g_{sub}, g, m)\times P^{E}(g_{sub}, g, m) .
$

The summation indicates that all possible subgraph matchings from $g_{sub}$ to $g$ should be considered when calculating the probability of $g_{sub}$ being sampled from $g$.

---

### Author Response · Authors · 2023-11-22
**Looking forward to your feedback**

Dear Reviewer,

Thank you for taking the time to review our paper. We appreciate your feedback and would like to address your concerns by revising the original paper, adding more experiments, and elaborating more on the technical details. We hope that our revisions address your concerns and provide a more comprehensive and compelling response.

We look forward to your feedback and hope to address any further questions or comments you may have!

Thank you.

---

### Author Response · Authors · 2023-11-23
**A kindly reminder**

Dear Reviewers,

Thank you for dedicating your time to review our work. Your feedback is highly valued, and we hope that our responses have successfully addressed all the issues you raised. In light of this, we humbly request that you reconsider your scoring. Moreover, as the period for open discussion is set to end in several hours, we kindly remind you to share any remaining concerns you might still have. We will be happy to provide further clarification!

Thank you so much for your consideration!

---

### Meta-Review · Area_Chair_8rDA · 2023-12-09

**Metareview:**

This paper studies interpretable graph networks with improved mutual information estimation. XAI in graph network is an important topic and this paper presents a way to improve current methods. However, the reviewers are concerns about the technical advances of this work. Thus a reject is recommended.

**Justification For Why Not Higher Score:**

The reviewers are concerns about the technical advances of this work.

**Justification For Why Not Lower Score:**

NA

---

### Decision · Program_Chairs · 2024-01-16

Reject